# Immunotherapy for Esophageal Cancer: State-of-the Art in 2021

**DOI:** 10.3390/cancers14030554

**Published:** 2022-01-22

**Authors:** Hugo Teixeira Farinha, Antonia Digklia, Dimitrios Schizas, Nicolas Demartines, Markus Schäfer, Styliani Mantziari

**Affiliations:** 1Department of Visceral Surgery, Faculty of Biology and Medicine UNIL, Lausanne University Hospital (CHUV), 1011 Lausanne, Switzerland; demartines@chuv.ch (N.D.); markus.schafer@chuv.ch (M.S.); styliani.mantziari@chuv.ch (S.M.); 2Department of Oncology, Faculty of Biology and Medicine UNIL, Lausanne University Hospital (CHUV), 1011 Lausanne, Switzerland; antonia.digklia@chuv.ch; 3First Department of Surgery, National and Kapodistrian University of Athens, Laikon General Hospital, 157 72 Athens, Greece; schizasad@gmail.com

**Keywords:** oesophageal cancer, tumor microenvironment, immunotherapy, esophageal adenocarcinoma, squamous cell cancer

## Abstract

**Simple Summary:**

The aim of this review was to describe the rationale for immunotherapy in different stages of esophageal cancer (EC) treatment, with a particular accent on curative intent treatment of locally advanced disease for the two predominant histological types (adenocarcinoma and squamous cell cancer). In addition to the already existing literature on immunotherapy for advanced and metastatic stages of EC, the current study provides a comprehensive review of the leading ongoing trials in 2021 with a focus on earlier stages of treatment in neo adjuvant and adjuvant settings.

**Abstract:**

The management of esophageal cancer (EC) has experienced manifold changes during the last decades. Centralization of EC treatment has been introduced in many countries, subsequently allowing the development of specialized high-volume centers. Minimal invasive surgery has replaced open surgery in many centers, whereas more potent systemic treatments have been introduced in clinical practice. Newer chemotherapy regimens increase long-term survival. Nevertheless, the overall survival of EC patients remains dismal for advanced tumor stages. In this direction, a wide range of targeted biologic agents (immunotherapy) is currently under assessment. Anti- Human Epidermal Growth Factor Receptor-2 (HER-2) monoclonal antibodies are used in HER2 (+) tumors, predominantly well-differentiated adenocarcinomas, and are currently assessed in the neoadjuvant setting (TRAP, INNOVATION trials). Immune checkpoint inhibitors Nivolumab (ATTRACTION-03) and pembrolizumab (KEYNOTE-181), have demonstrated a survival benefit compared with conventional chemotherapy in heavily pre-treated progressive disease. More recently, CheckMate-577 showed very promising results for nivolumab in a curative adjuvant setting, improving disease-free survival mainly for esophageal squamous cell carcinoma. Several ongoing trials are investigating novel targeted agents in the preoperative setting of locally advanced EC. In addition, other immunomodulatory approaches such as peptide vaccines and tumor infiltrating lymphocytes (TILs) are currently under development and should be increasingly integrated into clinical practice.

## 1. Introduction

### 1.1. Esophageal Cancer Treatment—Mixing Apples and Oranges

Esophageal cancer (EC) represents the 6th most frequent cancer-related cause of death worldwide, with 500,000 estimated new cases per year with an overall survival rate of 20% in five years [1,2]. The two predominant histological types are esophageal squamous cell carcinoma (ESCC) and esophageal adenocarcinoma (EAC). While ESSC represents more than 85% of all cases of EC worldwide, obesity and uncontrolled gastro–esophageal reflux lead to a significant increase of EAC. Indeed, the latter’s incidence and mortality has surpassed ESSC in several regions of the Western world [3,4,5], where ESCC tends to decrease or stabilize [6]. Although ESCC and EAC are increasingly recognized as two distinct diseases with distinct molecular patterns and risk factors [7], they have been treated similarly for many years [3]. Both are managed by surgical resection in early stages, with neoadjuvant chemo (radio-) therapy followed by surgery for locally advanced stages (cT3 and/or N+, M0) [3].

Following publication of the CROSS and FLOT trial results, a paradigm shift favors preoperative radio-chemotherapy for ESCC and perioperative chemotherapy for EAC [8,9]. However, even if the best available treatment is used, up to 30% of patients will present early recurrence within 12 months of surgery [10]. This is partly explained by the heterogeneity of response to standard (radio)-chemotherapy and the still limited understanding of individual tumor biology, which is not taken into account to adapt treatment options [11].

### 1.2. Understanding Immunity and Microenvironment of Esophageal Cancer; a Step towards Tailored Treatment

As in many solid tumors, a clear correlation has been observed between EC tumorigenesis and chronic pro-inflammatory conditions (tobacco and alcohol for ESCC, chronic gastroesophageal reflux and obesity for EAC), which induce high mutational rates [12,13]. The immune system is programmed to attack only foreign antigens, while recognizing autologous antigens as inoffensive. This response is coordinated by a balance between stimulatory and inhibitory immune signaling pathways [14], such as cytotoxic T-lymphocyte-associated antigen 4 (CTLA-4) and programmed cell death receptor 1 (PD-1) [15].

Cancer cells have the capacity to escape immunologic surveillance by disrupting the tumor microenvironment (TME), which is composed of immune cells, fibroblasts, endothelial cells or perivascular cells, and the extracellular matrix. Disruption of TME balance leads to tumor development by blocking apoptosis, allowing immune evasion and promoting angiogenesis, proliferation, and distant metastases [16].

Immunotherapy is a term used to describe all biologic/targeted agents that aim to increase and restore the immune system’s ability to detect and destroy cancer cells by modifying and/or blocking costimulatory signals [17,18,19,20]. This phenomenon seems to translate to a clinical benefit for patients, with phase II and III trials suggesting improved survival in esophageal and gastric cancer patients treated with immune checkpoint inhibition [4].

Immune checkpoint inhibitors have revolutionized cancer therapy, as a monotherapy or in various combinations; the three approved types are anti-PD-1, anti-PDL1, and anti-CTLA4 monoclonal antibodies [15]. Inhibition of activated CTLA-4, PD-1, and PD-L1 pathways can reverse regulatory T-cell-mediated immunosuppression [21].

Through overexpression of PD-L1 on the cancer cell surface or by inducing PD-L1 expression on the host’s immune cells, tumor cells use the PD-1/PD-L1 pathway to proliferate. When activated, PD-L1 has the ability to exhaust and inhibit host T-cell response that allows the tumor to escape immune surveillance [22]. As such, the PD-1/PD-L1 complex represents an ideal target for immunotherapeutic agents. The combined positive score (CPS) helps quantify the expression of PD-1/PD-L1 antigens and can be used to identify possible responders to anti-PD-1 therapy. A CPS ≥ 10 is considered PD-L1–positive and represents the percentage of PD-L1–expressing tumor and infiltrating immune cells within the total number of tumor cells [23]. Several molecules inhibiting the link between PD-1 and PD-L1 are currently used for the treatment of gastrointestinal tumors and specifically esophageal cancer: anti-PD-1 agents such as nivolumab and pembrolizumab, or anti-PD-L1-agents such as atezolizumab and avelumab [15].

CTLA-4 is a transmembrane protein, a homologue of the CD28 protein, and is expressed exclusively on activated T-cells. When CTLA-4 is bound to proteins, it prevents T cells from destroying other cells [19]. By inducing CTLA-4 upregulation on immune T-cells, cancer cells use the CTLA-4 pathway to escape and promote tumor growth [4,24,25]. Monoclonal antibodies inhibit upregulation of CTLA-4 in gastrointestinal cancer treatment, e.g., ipilimumab and tremelimumab [4].

Recent data have suggested that standard cytotoxic treatment has an impact on immune TME composition, influencing long-term prognosis in some tumors, such as pancreatic, rectal, and even EC [26,27,28,29,30,31]. This is where the role of targeted therapies seems to be most promising.

### 1.3. Mismatch Repair Protein Deficiency—Microsatellite Instability

Microsatellite instability (MSI) is characterized by a deficiency of DNA-mismatch repair proteins (dMMR), which results in accumulation of replication errors in DNA microsatellites (repeated DNA nucleotide sequences) [32]. The Human Genomic Atlas analysis identified a high prevalence of MSI-high (MSI-h) phenotype in gastric AC (22%), mostly in fundus and gastric body tumors [12]. A recent meta-analysis found that MSI-high gastric AC patients had a superior disease-free and overall survival compared to MSS (microsatellite stable) patients, whereas MSI-h status yielded a significantly worse survival using conventional cytotoxic chemotherapy [33]. However, two recent studies [34,35] report a variable prevalence of MSI-h status in EC, between 0 and 20% for EAC, and 0 and 60% for ESCC [35]. As PD-L1 antagonists are now a potential treatment option for MSI-h patients escaping standard chemotherapy [36,37], identification of these patients early in the course of the disease would be useful to adapt their systemic treatment.

As immunotherapy seems to be a promising treatment option for EC patients, a thorough understanding of its current indications and evidence is needed. Up to now, targeted agents have almost exclusively been used in a palliative setting if all other therapeutic options have failed. Its upfront use is only starting to be implemented in clinical practice. The aim of this review was to describe the rationale for immunotherapy in different stages of EC treatment, with a particular emphasis on curative intent treatment of locally advanced adenocarcinoma and squamous cell cancer. In addition to the already existing literature on immunotherapy for advanced and metastatic stages of EC, the current study provides a comprehensive review of the ongoing trials in 2021 with a focus on earlier stages of treatment in the neo adjuvant and adjuvant settings.

## 2. Materials and Methods

Four of the authors (HTF, AD, DS, SM) independently undertook electronic literature searches with Medline via PubMed; the detailed research strategy is shown in Appendix A. The references of the selected studies were hand-searched to identify relevant studies missed by the research algorithm. Ongoing trials on targeted therapy/immunotherapy in esophageal cancer (EC) were searched through www.clinicaltrials.gov, whereas the latest ASCO/ASCO GI and ESMO/ESMO GI congress abstracts were also reviewed for preliminary results of ongoing or recently finished trials.

Inclusion criteria were as follows: (i) comparative and non-comparative studies of targeted agents for esophageal cancer, for both histological types (adenocarcinoma, squamous cell cancer), (ii) studies with immunotherapy in a context of locally advanced or advanced/metastatic disease. Exclusion criteria were (i) trials involving various types of malignancy, including ‘gastroesophageal cancer’ without separate analysis for EC, (ii) registry studies with no specified treatment protocol, and (iii) studies with <10 patients.

## 3. Results

### 3.1. Immunotherapy in Esophageal Squamous Cell Cancer (ESCC)

#### 3.1.1. Advanced/Metastatic Setting

Programmed death-ligand 1 (PD-L1) expression can be found in up to 40–50% of ESCC [38]; targeted agents such as pembrolizumab [14] and nivolumab [15] have shown promising results in ESCC [38,39].

The KEYNOTE-028 trial (multicenter, randomized phase Ib) treated PD-L1-positive EC patients with pembrolizumab [40], whereby 78% were ESSC. Pembrolizumab showed a prolonged antitumor activity without any relevant toxicity. Median progression-free survival (PFS) was 1.8 months (95% CI, 1.7 to 2.9), and median OS was 7 months (95% CI, 4.3 to 17.7).

The KEYNOTE-590 trial compared pembrolizumab and chemotherapy (cisplatin/5-FU) versus chemotherapy alone in 273 patients with locally advanced/unresectable or metastatic including ESCC (73%) [41]. Pembrolizumab and chemotherapy offered superior OS in ESCC patients (with PD-L1 CPS ≥ 10), with a median of 13.9 versus 8.8 months (HR 0.57 [95% CI, 0.43 to 0.75]; *p* < 0.0001) respectively. Both treatment arms presented an acceptable safety profile [41]. Comparable findings were published in the KEYNOTE-181 phase III trial for advanced ESCC with a CPS >10, treated with second-line pembrolizumab versus chemotherapy. Median OS was 8.2 months for pembrolizumab arm versus 7.1 months (HR, 0.78 [95% CI, 0.63 to 0.96]; *p* = 0.0095) [42].

Nivolumab was assessed in the ATTRACTION-1 phase II trial [43] in 65 ESCC patients refractory or intolerant to platinum, taxane, and fluoropyrimidine chemotherapy. Nivolumab showed a promising safety profile, and tumor load and target lesions decreased in 29 patients (45%). Median PFS was 1.5 months (95% CI, 1.4 months to 2.8 months), and median OS was 10.8 months (95% CI, 7.4 months to 13.3 months). Comparable outcomes were published in the ATTRACTION-3 phase III trial; 208 ESCC patients were assigned to the nivolumab arm and 209 to the standard chemotherapy arm. Nivolumab was associated with a significant improvement in OS with a median of 10.9 months (95% CI, 9.2 months to 13.3 months) versus 8.4 months for chemotherapy (95% CI, 7.2 months to 9.9 months) (*p* = 0.019) with favorable safety profile for patients with advanced ESCC [44].

#### 3.1.2. Adjuvant Setting

Neoadjuvant radio-chemotherapy (RCT) is the standard treatment for patients with ESCC in several parts of the world based on the results of the landmark CROSS trial [3,9,38]. However, some patient treated with neoadjuvant RCT would also require additional adjuvant oncological treatment. Such adjuvant therapies are not yet standardized after esophagectomy.

The recently published phase III CheckMate 577 trial compared adjuvant nivolumab to placebo in 794 patients with resected (R0) stage II or III EC or GECJ after neoadjuvant RCT [45]. Nivolumab was administered at a dose of 240 mg/2 weeks for 16 weeks, then 480 mg/4 weeks, for a total treatment time up to one year. The histological characteristics of included patients were as follows: 60% EC (29% ESCC and 71% EAC) and 40% of the GECJ. Patients with complete pathological response were excluded. After a median follow-up of 24.4 months, DSF was significantly increased in the nivolumab arm (22.4 months (95% CI, 16.6 months to 34 months). The benefit of nivolumab was found in all subgroups (EC and GECJ) but appeared to be greater for ESCC with a median of DFS of 29.7 months (HR 0.61 [95% CI, 0.43 to 0.75]). The safety profile was good with grade 3–4 side effect rates of 34% in the nivolumab arm and 32% in the placebo arm. Thus, after a median follow-up of two years, nivolumab was associated with a reduction in the risk of recurrence or death by 31% [45]. A post hoc analysis showed a DFS benefit of nivolumab (HR, <1) in patients with a ≥5 CPS.

### 3.2. Immunotherapy in Esophageal Adenocarcinoma (EAC)

#### 3.2.1. Advanced/Metastatic Setting

In EAC and GEJC patients, response to immunotherapy seems to be dependent on the PD-L1 CPS. Over-expression of PD-L1 (CPS ≥ 10). was associated with better pathological response and overall survival [38]. As in ESCC, PD-L1 expression has been described in 40% of EAC and was found to be higher in the Microsatellite instability (MSI) subtype [38]. In the KEYNOTE-028 trial, the overall response rate in EAC patients was 40% with pembrolizumab [40]. In the CheckMate-032 study (multicenter, randomized phase I), where 63% (101/160) of patients had advanced metastatic EAC or GEJC [46], overall response to nivolumab was reported in 12% of patients; this increased to 24% when combined with ipilimumab, but with higher rates of toxicity.

As a first palliative line for advanced junction and lower EAC, the CheckMate-649 study (randomized, phase III) demonstrated improved OS (13.8 months versus 11.1 months) for patients with CPS ≥ 5 for nivolumab + chemotherapy (HR 0.71 [98.4% CI 0.59 to 0.86]; *p* < 0.0001) [47]. In the KEYNOTE-590 trial, pembrolizumab was studied in combination with chemotherapy in advanced/unresectable GEJ adenocarcinomas, with satisfactory response rates and acceptable toxicity [41].

#### 3.2.2. Adjuvant Setting

The CheckMate-577 study is the only published randomized trial on adjuvant immunotherapy, including 244 patients with EAC [45]. A significant benefit in DFS was observed in patients treated with nivolumab versus placebo in the overall group and for each histological type, irrespective of the lymph node status and PD-L1 status. DFS for EAC patients was 19.4 months for the nivolumab arm versus 11 months for placebo (HR 0.75, 95% CI 0.42 to 0.88, *p* < 0.001). Of note, no data for overall survival were presented with a follow up of 24 months [45].

An overview of the published trials regarding Advanced/metastatic and adjuvant settings for ESCC and EAC is presented in Table 1.

### 3.3. Immunotherapy in EC; Ongoing Trials in 2021 and Preliminary Results

An overview of the 40 ongoing trials and the 14 recently published trials with preliminary data are presented in Table 2 and Table 3.

#### 3.3.1. Neoadjuvant Setting

##### Anti-Human Epidermal Receptor-2 (HER-2) Targeted Therapy

Preliminary results of the NRG Oncology/RTOG 1010 phase III randomized trial (NCT01196390) on patients with locally advanced HER-2 positive EAC were recently reported. The addition of trastuzumab to neoadjuvant chemoradiation (carboplatin/paclitaxel + 50.4Gy) did not provide any significant DFS (19.6 month trastuzumab versus 14.2 month control, *p* = 0.85) or OS (38.5 month trastuzumab versus 38.9 month control) benefit to the targeted therapy group. Recently, the TRAP trial demonstrated 34% pCR rates and an improved overall survival in HER2 (+) EAC patients treated with chemoradiation and a combination of trastuzumab/pertuzumab [49]. The INNOVATION EORTC-1203-GITCG trial [50] investigates the combination of standard chemotherapy with trastuzumab/pertuzumab in HER-2 overexpressing gastroesophageal adenocarcinoma in the neoadjuvant setting. Similarly, the MATTERHORN (NCT04592913) and MONEO (NCT03979131) trials are assessing the combination of perioperative FLOT [8] and darvolumab and atelumab, respectively, in gastric and gastroesophageal junction cancer.

##### PD-1/PD-L1 Immune Checkpoint Inhibitors

Several ongoing trials are investigating novel PD-1/PD-L1 checkpoint inhibitors in the preoperative setting of locally advanced EC.

Nivolumab in association with neoadjuvant chemotherapy (cisplatin/docetaxel/5FU) is being assessed in a phase I trial in ESCC/EAC (NCT03914443), and as a single agent for ESCC patients (three cycles preoperatively) in a phase II study (NCT03987815). Preliminary results of the FRONTiER trial (NCT03914443) suggest a rate of >50% serious adverse effects (grade 3–4) when nivolumab was added to 5FU/cisplatin, with a 33.3% pathologic complete response (pCR).

Pembrolizumab in association with CROSS-protocol (carboplatin/paclitaxel +41.4Gy, [9] chemoradiation is under assessment in two ESCC phase I/II trials (NCT04435197, NCT03792347). Nivo- and pembrolizumab are also under phase I-II assessment in combination with other targeted agents (cetuximab, lenvatinib, relatlimab) (Table 2).

Tislelizumab in association with [9] chemoradiation or carboplatin/paclitaxel chemotherapy is being studied in three phase II–III trials (NCT04973306, NCT04974047, NCT04776590), all in Asian populations of ESCC.

Another recently approved checkpoint inhibitor in China, toripalimab is under assessment in several phase II studies for locally advanced ESCC, with either preoperative chemotherapy (NCT03985670, NCT04177797, NCT04804696) or chemoradiation with the CROSS regimen (NCT04888403, NCT04177875, NCT04006041, NCT04644250). One phase III randomized trial, investigating preoperative paclitaxel/cisplatin with or without toripalimab, is expected to include 500 patients until May 2026 (NCT04848753). Preliminary results from the NCT03985670 trial report a 36.4% pCR after sequential administration of toripalimab (two days after the start of cisplatin/taxane chemotherapy).

Anti-PD-1 blockade with camrelizumab is being extensively studied in Asian populations of locally advanced ESCC. Three phase I-II trials with camrelizumab added to neoadjuvant platin/taxane regimens are ongoing (NCT03917966, NCT04506138, NCT04767295), whereas preliminary results reported pCR rates between 25–42.5% and 37% serious adverse events after platin/taxane and camrelizumab neoadjuvant therapy (ChiCTR2000028900, NCT04225364, ChiCTR1900026240, NCT03917966, ChiCTR1900023880). Sintilimab, durvalumab, SHR-131 and socazolimab are among other recent checkpoint inhibitors in the course of phase I–II assessment in ESCC patients (Table 2). The only preliminary results available report a 25% pCR for sintilimab in a phase II trial (ChiCTR1900026593).

##### Other Targeted Agents

Anti-VEGF agents (bevacizumab) and other tyrosine-kinase inhibitors (apatinib, sotigalimab) are in preliminary (phase I–II) stages of assessment in the neoadjuvant setting of EC.

#### 3.3.2. Definitive Chemoradiation

Two phase III randomized trials are currently assessing nimotuzumab (NCT02409186) and tislelizumab (NCT03957590) combined with definitive, potentially curative, chemoradiation (cisplatin/paclitaxel + 50–59.4Gy) for ESCC patients.

#### 3.3.3. Perioperative- Adjuvant Treatment

Targeted therapy is an area of intense interest in the perioperative setting for locally advanced, resectable EC. The KEYSTONE 1 and 2 trials (NCT04389177, NCT04807673) are currently evaluating the benefit of perioperative pembrolizumab in pCR rates and DFS in an Asian ESCC population. A large phase III randomized trial assesses toripalimab in ESCC patients (NCT04280822).

Two phase II trials are investigating perioperative checkpoint inhibitors (sintilimab, avelumab) in EAC patients (NCT04989985, NCT03490292), whereas preliminary phase I results of the NCT03604991 trial reported no additional toxicity when perioperative nivolumab was added to the standard CROSS regimen.

Darvolumab has shown some promising preliminary results in the perioperative and adjuvant settings in EAC patients, with 24% pCR and 12.3% serious adverse effects (grade 3/4). (NCT02962063, NCT02639065). The combination of PD-L1 inhibitors to the FLOT regimen [8] was recently proposed. Al-Batran, in an interim safety analysis of the DANTE trial (NCT03421288), reported high rates of toxicity (80% in the arm FLOT-atezolizumab versus 70% in the FLOT arm), although a proportion was attributed to preoperative comorbidity; similarly, interim results of the ICONIC trial (NCT03399071) suggest 60% grade 3–4 toxicity after FLOT/avelumab [51].

## 4. Discussion

In recent years, immunotherapy has demonstrated promising results as part of the armamentarium of EC treatment. Immune checkpoint inhibitors (PD-1/PD-L1 blockade) were firstly used in advanced or metastatic disease, in heavily pre-treated patients. After the recent CheckMate-577 trial suggesting a significant benefit of adjuvant nivolumab after standard CROSS regimen, several ongoing studies are assessing checkpoint inhibitors in the perioperative context. This places immunotherapy studies in the curative setting, which would be a major step forward for esophageal cancer. However, there are various issues that need particular attention before integrating immunotherapy on a large scale.

### 4.1. Future Perspectives and Challenges in Targeted Therapy of EC

#### 4.1.1. Tumor Microenvironment (TME); the Tumor’s Signature and the Key to Targeted Treatment

The tumor-infiltrating lymphocytes (TILs), as part of the host’s defense mechanism against solid tumors, play an important role. Each tumor triggers an individual immunologic response with a large variability in the number and type of TILs that form its specific TME. The presence and density of TILs have been correlated with better long-term prognosis and improved response to immunotherapy [52]. Many different types of lymphocytes have been identified in the TME (CD4+ with/without suppressor FoxP3+ expression, CD8+ with/without PD-L1 expression, and M2 macrophages) [53,54], reflecting the complex interplay between tumor antigenicity and host immune reaction. Increased CD8+ infiltration of the stroma and tumor margins has been associated with better OS and DFS in EC patients [28,29,30,55,56], and in particular, in EAC [57]. Conversely, CD4+ cells expressing the forkhead box transcription factor (FoxP3+, 5–10% of all CD4 lymphocytes) have been associated with a local immunosuppressive effect, enhancing immunotolerance against solid tumors [58,59]. However, the exact prognostic value of the TME is not yet clear for EC, and it has few clinical implications up to this day.

Recently, pre-treatment M2 macrophage infiltration was associated with poor response to chemotherapy and shorter DFS in ESCC patients [53,54], whereas the presence of FoxP3+ TILs was also related to a worse prognosis in ovarian cancer patients [58,60]. On the other hand, chemotherapy and radiation have been proven to modify pre-treatment TME; although the exact mechanism has yet to be elucidated, cellular destruction by cytotoxic treatment leads to larger exposure of the intra-tumoral mutational load and elicits local cytokine production, stimulating the host’s immune system [61]. The immune response is enhanced mostly by increased peritumoral CD8+ TIL and/or suppression of inhibitory Treg (FoxP3+) cells [58,61]. According to previous studies on rectal cancer, external beam radiation induces a significant decrease of inhibitory (FoxP3+) cells and modifies the CD8+/FoxP3+ ratio of TILs in the tumor and stromal tissue, leading to improved progression-free survival [26,58].

The cellular TME, in addition to its unique immunologic signature, has a therapeutic potential. Tumor lymphocytes (TILs) activated against tumor antigens are retrieved and genetically engineered (cloned) in vitro before being re-infused in the patient [15]. This innovative line of treatment has shown promising results in metastatic melanoma patients, but also in other types of solid tumors with poor prognosis and limited therapeutic options (e.g., cholangiocarcinoma) [62,63]. Currently, NeoTIL-ACT is an ongoing phase I pilot trial for patients with recurrent or metastatic solid tumors (NCT04643574), based on lymphodepleting chemotherapy followed by with low-dose radiation and infusion of autologous expanded TILs enriched for tumor antigen specificity (NeoTIL). Although the study was only recently initiated (2021), it is very promising in the context of advanced, metastatic disease in heavily pre-treated oncologic patients.

#### 4.1.2. Future Outlook: Cancer Vaccines and CAR T-Cell Therapy

Cancer vaccines have been for a long time one of the high hopes of humanity to cure cancer. However, due to the variable antigenicity and significant mutational load even among tumors of the same type, development of efficient cancer vaccines has remained a challenge [38,64].

Dendritic cells (DCs), as part of the host’s immune response, have a great capacity to present antigens on their surface, triggering intense cytotoxic lymphocyte response [65]. DC vaccines were recently tested in a randomized study of 40 EC patients undergoing radiation and surgery [66]. Tumor heat-shock proteins were extracted from surgical specimens and cultured in vitro with the patient’s autologous DC and these were then re-infused to the patient. The experimental vaccine group not only showed a significantly increased immune response (higher circulating IL-2, IL-12 and INF-γ, and CD8+ cytotoxic cells) but also a better 2-year survival compared to patients treated only with radiation and surgery. However promising, these results are preliminary and need further validation before implementing in clinical practice.

Similarly, peptide vaccines, created from antigens retrieved in tumor lysate, aim to stimulate and activate the host’s cytotoxic T cells, reinforcing innate antitumor activity [64,67]. Recent data suggest a significantly enhanced immunological response in ESCC patients treated with peptide vaccine preoperatively (e.g., NY-ESO-1, S-588410, other multi-peptide vaccines), with an acceptable safety profile (e.g., NY-ESO-1, S-588410, other multi-peptide vaccines) [68]. Promising 5-year results of another multi-peptide vaccine tested have just been published, when administered to lymph-node positive ESCC patients as an adjuvant after surgery [69]. In this non-randomized phase II trial, cancer-free survival was significantly better in patients treated with the vaccine after surgery [69]. Several ongoing trials are currently evaluating peptide vaccines in adjuvant and metastatic settings for ESCC (NTC01697527, NCT00995358) [38].

CAR T-cell therapy is another potent immunotherapeutic treatment approach, aiming to genetically engineer natural killer T cells to target specific tumor antigens [15]. Although promising results have been shown in hematologic malignancies, the solid tumor environment is less favorable for these agents. This is due to the lack of tumor-specific antigens, the potential local immunosuppressive action from the activated PD-1/PD-L1 complex, as well as the severe reported toxicities resembling a systemic cytokine storm [70].

#### 4.1.3. Geographical Differences in EC Treatment

The role of immunotherapy has mostly been proven for ESCC. The vast majority of these trials are conducted in Asian populations with a high prevalence of ESCC lesions. However, little is known as to whether these results can be safely extrapolated to the Western population where EAC is the prevalent form of EC. It has previously been proven that even for similar histologic types of cancer, such as gastric adenocarcinoma, there are inherent differences in biologic behavior and prognosis between Eastern and Western populations, with superior survival in patients from Asian series [71,72,73,74]. In gastric cancer, inherent differences in tumor biology seem to play a role, as these differences remain even when tumor stage, perioperative treatment, and extent of lymphadenectomy are accounted for [71,72].

Therefore, how certain are we that EC with its inherent histologic polymorphism will be as responsive to targeted agents for Western patients as it seems to be in Eastern ESCC populations? As seen above, the Asian series have the lion’s share in ongoing clinical trials on immunotherapy for EC. This might introduce significant bias when trying to implement the same therapies to western, EAC-predominant, series.

#### 4.1.4. Health-Policy Issues

An often neglected aspect of novel therapies is the methodological, legal, and ethical framework that determines their research and development. Currently, the overwhelming majority of scientific research on immunotherapy is industry-driven, with few or no measures implemented to guarantee the absence of data dredging or publication bias. International study registries exist to enhance transparency in medical research, but most of these studies are then published even when major discrepancies are seen between the study protocol and the reported outcomes. Thus, instead of multiplying ‘feasibility’ trials whose ethical regulations and anticipated benefit remain obscure, the scientific community and related stakeholders should focus on guaranteeing the reliability, quality, and expected clinical benefit of ongoing and future trials on this fascinating field of immunotherapy [75].

Last but not least, the financial aspect of all these novel treatments needs to be considered. Nivolumab is currently being introduced to a large EC patient population following the results of the hallmark CheckMate-577 study [45]. Taking a closer look at the trials, although nivolumab offered a robust benefit in terms of progression-free survival, which was the primary endpoint, OS data are still pending, but it is OS benefit that drives insurance reimbursement worldwide. Nivolumab is currently approved by the EMA (European Medicines Agency, Amsterdam, The Netherlands) and the FDA (Food and Drug Administration, White Oak, MD, USA). It is also accepted and reimbursed upon demand by health insurances in Switzerland, where the financial cost per patient/year is estimated at €100.000. Previous USA-based analyses estimated the cost to be even higher, at $6,676 per cycle, thus $160.160 a year [76]. However, little is known about the actual financial burden (actual price, adverse events, post-progression treatment) it represents and its cost-effectiveness.

Recent data from the field of hepatocellular carcinoma compared pembrolizumab to placebo as a second-line treatment showed an incremental 0.153 life-year benefit for the anti-PD-1 agent, with a supplementary cost of $47.057 per year [77]. In this context, it was estimated that either a survival benefit >12 months or a significant reduction in price (57.7%) are needed for it to become cost-effective [77]. As pembrolizumab and nivolumab have comparable financial costs [76], caution is needed before implementing this treatment on a large scale without taking into account its financial aspect.

Health care providers and all related policy makers need to be conscious of the considerable financial stakes related to immunotherapy. Health care systems around the world need to be able to afford these treatments without jeopardizing the already fragile financial balance most of them face, and most importantly, without compromising equity of patient access to care.

## 5. Conclusions

Esophageal cancer treatment enters a new era where targeted immunotherapeutic agents are increasingly used to complement or even replace classic cytotoxic agents. An overwhelming number of studies are currently ongoing to assess all these novel agents, with the aim to stimulate and specifically drive the host’s immune system against cancer antigens. Immunotherapy is a complex, fascinating, and potentially practice-changing field of cancer research, with the list of novel treatment lines growing exponentially (immune checkpoint inhibitors, adoptive TILs, cancer vaccines, CAR T-cell therapies). Although promising preliminary results have been published, large-scale studies and long-term results from both a clinical and financial point of view must be mandated before wide implementation in clinical practice.

## Figures and Tables

**Table 1 cancers-14-00554-t001:** Summary of published results.

Réf	Trial	Target	Phase	N	Histology	Arm 1	Arm 2	Primary Endpoint	Secondary Endpoint
**Advanced/Metastatic Treatment**									
[40]	KEYNOTE-028	PD-1/PD-L1	Ib	23	78% ESSC	Pembrolizumab alone	-	ORR 30% in PD-L1	PFS 1.8 months (95% CI, 1.7 to 2.9 months) OS 7 months (95% CI, 4.3–17.7 months)
[41]	KEYNOTE-590	PD-1/PD-L1	III	372	73% ESCC (*n* = 273) 27% EAC (*n* = 99)	Pembrolizumab + chemotherapy (Cisplatin/5-FU)	Chemotherapy alone	OS for ESCC (CPS ≥ 10) 13.9 versus 8.8 months HR 0.57 [95% CI, 0.43–0.75]	
[42]	KEYNOTE-181	PD-1/PD-L1	III	314	63% ESCC (*n* = 198) 37% EAC (*n* = 116)	Pembrolizumab	Chemotherapy	OS 8.2 versus 7.1 months HR, 0.78 [95% CI, 0.63–0.96]	
[43]	ATTRACTION-1	PD-1/PD-L1	II	65	ESCC	Nivolumab	-	OS 10.8 months (95% CI, 7.4–13.9) 5-year OS 6.3% (95% CI, 2.0–14.0)	PFS 1.5 months (95% CI, 1.4–2.8) 5-year PFS 6.8% (95% CI, 2.2–15.1)
[44]	ATTRACTION-3	PD-1/PD-L1	III	208	ESCC	Nivolumab	Chemotherapy (Placitaxel/ Docetaxel)	OS 10.9 months (95% CI, 9.2–13.3 months) versus 8.4 months (95% CI, 7.2–9.9 months)	
[46]	CheckMate-032	PD-1/PD-L1/ CTLA-4	I	160	63% EAC (*n* = 101)	Nivolumab	Nivolumab + Ipilimumab	ORR 12% versus 24%	
[47]	CheckMate-649	PD-1/PD-L1	III	789	13% EAC (*n* = 103)	Nivolumab + Chemotherapy	Chemotherapy alone	OS (CPS ≥ 5) 13.8 versus 11.1 months (HR 0.71, 98·4% CI 0.59–0.86)	
**Adjuvant Treatment**									
[45]	CheckMate-577	PD-1/PD-L1	III	531	29% ESCC (*n* = 155) 71% EAC (*n* = 376)	Adjuvant Nivolumab	Placebo	DFS for ESCC 29.7 months versus 11 months (HR 0.61, 95% CI, 0.43–0.75) DFS for EAC 19.4 months for versus 11 months (HR 0.75 95% CI 0.42–0.88)	

DFS = disease-free survival, PFS = progression-free survival, EAC = esophageal adenocarcinoma, ESCC = esophageal squamous cell cancer, OS = Overall Survival, ORR = Objective Response Rate per RECIST 1.1.

**Table 2 cancers-14-00554-t002:** Ongoing trials.

	Study Identifier- Country	Estimated Start-End Date	Nb of Patients	Inclusion Criteria	Study Design	Arm 1	Arm 2	Primary Endpoint	Status
**Neoadjuvant Treatment**									
	**(a) PD-1/PD-L1 Checkpoint Inhibitors**								
1.	NCT03914443 Japan	7 May 2019– 1 February 2022	36	ESCC, AC	Phase I	Cisplatin/5FU/Docetaxel Nivolumab	-	Toxicity	Active, not recruiting
2.	NCT03987815 Korea	1 August 2019– 31 July 2022	20	ESCC	Phase II	Nivolumab (3 cycles)	-	Major pathologic response	Recruiting
3.	NCT03792347 China	21 January 2019– 17 June 2020	20	ESCC	Phase I	CROSS * Pembrolizumab	-	Toxicity	Active, not recruiting
4.	NCT04435197 China	11 August 2020– June 2025	143	ESCC	Phase II	CROSS * Pembrolizumab	-	pCR	Recruiting
5.	NCT04973306 China	July 2021– July 2027	176	EC	Phase II/III RCT	CROSS * Tislelizumab	CROSS *	pCR	Not yet recruiting
6.	NCT04974047 China	17 August 2021– 30 May 2026	65	ESCC	Phase II Non-randomized	Carboplatin/Paclitaxel Tislelizumab	Carbotaxol/Paclitaxel Tislelizumab Chemoradiation	pCR	Recruiting
7.	NCT04776590 China	28 January 2021– 15 December 2024	30	ESCC	Phase II	CROSS * Tislelizumab		pCR	Recruiting
8.	NCT04848753 China	12 May 2021– 12 May 2026	500	ESCC	Phase III RCT	Cisplatin/Paclitaxel Toripalimab	Cisplatin/Paclitaxel	Event-free survival	Recruiting
9.	NCT04006041 China	25 June 2019– 21 December 2020	44	ESCC	Phase II	Cisplatin/Paclitaxel/44 Gy Toripalimab	-	pCR	Recruiting
10.	NCT04644250 China	1 September 2020– 1 March 2024	32	ESCC	Phase II	CROSS * Toripalimab	-	pCR	Recruiting
11.	NCT04177797 China	20 March 2020– 31 December 2021	20	ESCC	Phase II	Carboplatin/Paclitaxel Toripalimab	-	pCR	Active, not recruiting
12.	NCT04804696 China	April 2021– March 2024	53	ESCC	Phase II	Carboplatin/Paclitaxel Toripalimab	-	pCR	Recruiting
13.	NCT04888403 China	3 December 2021– 2 July 2022	45	ESCC	Phase II	Paclitaxel/Nedaplatin 41.4Gy Toripalimab	.	pCR	Not yet recruiting
14.	NCT04177875 China	1 May 2019– 30 April 2022	40	ESCC	Phase II	Docetaxel/Paclitaxel/Cisplatin + 40Gy Toripalimab	-	Major pathologic response	Recruiting
15.	NCT03917966 China	7 April 2020– October 2022	60	ESCC	Phase II	Docetaxel/Nedaplatin Camrelizumab	-	ORR	Recruiting
16.	NCT04506138 China	11 August 2020– 21 December 2025	46	ESCC	Phase I/II	Carboplatin/Paclitaxel Camrelizumab	-	Major Pathologic response	Recruiting
17.	NCT04767295 China	1 March 2021– 1 March 2023	28	ESCC	Phase II	Carboplatin/Paclitaxel Camrelizumab	-	ORR	Recruiting
18.	NCT04625543 China	December 2020– September 2023	100	ESCC (PD-L1 > 10%)	Phase II RCT	Paclitaxel/Cisplatin Sintilimab	Paclitaxel/Cisplatin	Major pathologic response	Not yet recruiting
19.	NCT03946969 China	8 May 2019– 1 October 2022	40	ESCC	Phase I-II	Cisplatin/Paclitaxel/S-1 Sintilimab	-	Toxicity	Recruiting
20.	NCT04568200 China	19 June 2020– December 2023	60	ESCC	Phase II Non-randomized	CROSS * Durvalumab	CROSS *	Pathologic response	Recruiting
21.	NCT04215471 China	February 2020–December 2020	30	ESCC	Phase II	SHR-131	-	ORR	Not yet recruiting
22.	NCT04460066 China	1 November 2020– 1 November 2023	70	ESCC	Phase I/II RCT	Paclitaxel/Cisplatin Socazolimab	Paclitaxel/Cisplatin Placebo	Major pathologic response	Not yet recruiting
	**(b) Combined Therapy**								
23.	NCT04229459 Israel	30 December 2019– June 2027	31	ESCC	Phase I/II	5FU/Cisplatin +50.4 Gy Cetuximab Nivolumab	-	pCR	Recruiting
24.	NCT04929392 USA	1 October 2021– 3 December 2023	24	ESCC, EAC	Phase II	CROSS * Pembrolizumab Lenvatinib	-	pCR	Recruiting
25.	NCT03044613 USA	11 July 2017– February 2024	32	ESCC, EAC	Phase IB Non-randomized	CROSS * Nivolumab	CROSS * Nivolumab Relatlimab	Treatment-related toxicity	Active, not recruiting
	**(c) Other Targeted Therapy**								
26.	NCT02812641 Taiwan	June 2016– December 2021	50	Stage III ESCC	Phase I/II RCT	5FU/Cisplatin 40Gy Bevacizumab	5FU/Cisplatin 40Gy	pCR	Recruiting
27.	NCT03165994 USA	6 October 2017– 31 December 2021	26	ESCC, EAC	Phase I/II	Carboplatin/Paclitaxel 50.4Gy Sotigalimab(APX005M)	-	Treatment-related toxicity	Recruiting
28.	NCT03857763 China	1 March 2019– 1 March 2023	40	ESCC	Phase II	CROSS * Apatinib	-	pCR	Not yet recruiting
**Definitive Chemoradiation**									
29.	NCT02409186 China	March 2015– December 2021	200	ESCC	Phase III RCT	Cisplatin/Paclitaxel 59.4Gy Nimotuzumab	Cisplatin/Paclitaxel 59.4Gy Placebo	OS	Recruiting
30.	NCT03957590 China	12 June 2019– 30 October 2023	316	ESCC	Phase III RCT	Cisplatin/Paclitaxel 50.4Gy Tislelizumab	Cisplatin/Paclitaxel 50.4Gy Placebo	PFS	Recruiting
**Perioperative Immunotherapy**									
31.	NCT04389177 KEYSTONE 001, China	8 July 2020– 31 December 2024	50	ESCC	Phase II RCT	Cisplatin/Paclitaxel Perioperative Pembrolizumab	-	Major pathologic response	Recruiting
32.	NCT04807673 KEYSTONE-002, China	May 2021– May 2028	342	ESCC	Phase III RCT	CROSS * Perioperative Pembrolizumab	Paclitaxel/Cisplatin Pembrolizumab Surgery	Event-free survival	Recruiting
33.	NCT02844075 China	January 2017- May 2022	18	ESCC	Phase II	CROSS * Perioperative Pembrolizumab	-	pCR	Active, not recruiting
34.	NCT04437212 China	1 July 2020– 3 December 2023	20	ESCC	Phase II	CROSS * Perioperative Toripalimab	-	Major pathologic response	Recruiting
35.	NCT04280822 China	21 April 2020– 2 March 2028	400	ESCC	Phase III RCT	Cisplatin/Paclitaxel Perioperative Toripalimab	Cisplatin/Paclitaxel Surgery	Event-free survival	Recruiting
36.	NCT04989985 China	1 September 2021– 1 August 2027	302	Junction EAC	Phase II RCT	Perioperative Oxaliplatin/S-1 Sinitilimab	Perioperative Oxaliplatin/S-1	pCR	Recruiting
37.	NCT03490292 USA	29 May 2018– February 2024	24	ESCC, EAC	Phase I/II	CROSS * Perioperative Avelumab	-	Toxicity	Recruiting
**Adjuvant Treatment**									
38.	NCT04159974 Germany	30 September 2019– June 2024	56	EAC	Phase II RCT	Chemoradiation Durvalumab Surgery	Chemoradiation Surgery Durvalumab/ Tremelimumab	pCR	Recruiting
39.	NCT02520453 Korea	February 2016– December 2021	86	ESCC	Phase II RCT	Durvalumab	Placebo	DFS	Active, not recruiting
40.	ChiCTR2100045651 China	May 2021– December 2022	220	ESSC	Phase III	Cisplatin-based doublet Tislelizumab	Tislelizumab	DFS	Active

DFS = disease-free survival, EAC = esophageal adenocarcinoma, ESCC = esophageal squamous cell cancer, HER-2 = Human Epidermal Receptor-2, pCR = pathologic complete response, RCT = Randomized Controlled Trial, OS = Overall Survival, ORR = Objective Response Rate per RECIST 1.1. CROSS * = Carboplatin/Paclitaxel + 41.4Gy according to the CROSS regimen [9]. Major pathologic response = TRG1-2 according to Mandard [48].

**Table 3 cancers-14-00554-t003:** Preliminary results summary.

Principal Investigator Study Identifier	Nb of Patients	Inclusion Criteria	Study Design	Treatment Details	Primary Endpoint	Primary Endpoint Results
**Neoadjuvant Treatment**						
Cheng, Chao ChiCTR2000028900	20	ESCC	Phase II	Carboplatin/Paclitaxel Camrelizumab	pCR	27.8%
Li, Jingpei NCT04225364	56	ESCC	Phase II	Nabpaclitaxel/Cisplatin Camrelizumab	pCR	35.3%
Li, Zhigang ChiCTR1900026240	60	ESCC	Phase II	Carboplatin/Paclitaxel Camrelizumab	pCR	42.5%
Wang, Feng NCT03917966	26	ESCC	Phase II	Nedaplatin/Docetaxel Camrelizumab	Major pathologic response, pCR	42% major response 25% pCR
Wang, Zhen ChiCTR1900023880	30	ESCC	Phase Ib	Chemotherapy (Nabpaclitaxel/Platin/Apatinib) Camrelizumab	Safety and feasibility	80% patients received all planned cycles, 36.7% serious adverse effects
Zhao, Lingdi NCT 03985670	30	ESCC	Phase II	Simultaneous versus sequential chemo-immunotherapy (Paclitaxel/Cisplatin+ Toripalimab)	pCR	36.4% sequential versus 7.7% simultaneous, *p* = 0.079
Safran, Howard NCT01196390	203	EAC, HER2 (+)	Phase III, RCT	Carboplatin/Paclitaxel + 50.4Gy +/− Transtuzumab	DFS OS	Median DFS: 19.6 mo (CR/T) versus 14.2 mo (CR), *p* = 0.85
Yamamoto, Shun NCT03914443 FRONTiER	13	ESCC	Phase I	5FU/Cisplatin Nivolumab	toxicity pCR	50% ≥ grade 3 adverse events 33.3% pCR
Zhang Z ChiCTR1900026593	40	ESCC	Phase II	Carboplatin/Paclitaxel Sintilimab	Major pathologic response	47.5% major response 25% pCR
**Perioperative-Adjuvant Treatment**						
Al-Batran SE NCT03421288 DANTE trial	40	Gastro-EAC	Phase II	FLOT [8] Perioperative Atezolizumab + adjuvant Atezolizumab	Adverse events	80% in arm FLOT-A, 70% in arm FLOT
Eads, Jennifer NCT03604991	31	EAC	Phase I RCT	Carboplatin/Paclitaxel + 41.4Gy Perioperative Nivolumab	Safety, side effects	No disproportionate toxicity added by Nivolumab
YuyatKu, Geoffrey NCT02962063	36	EAC	Phase I/II	5FU/platin + 50.4Gy Perioperative Darvolumab	pCR	pCR 24%
Mamdani Hirva NCT02639065	24	EAC	Phase II	5FU/Cisplatin + radiation Adjuvant Darvolumab	Toxicity	12.3% ≥ grade 3 adverse events
Athauda, Avani NCT03399071 ICONIC	15	EAC	Phase II/I	FLOT [8] Perioperative Avelumab	Treatment-related toxicity R0 rate	60% Grade 3–4 toxicity 100% R0

ESCC = Esophageal Squamous cell carcinoma, EAC = esophageal adenocarcinoma, RCT = Randomized Controlled Trial. pCR = pathologic complete response.

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
