# Peer review of "Immunotherapy for Esophageal Cancer: State-of-the Art in 2021"

_cancers, 2022, doi:10.3390/cancers14030554_

Round 1

Reviewer 1 Report

The review manuscript is aimed at describing the rationale for immunotherapy in different stages of esophageal cancer treatment, with a particular accent on curative intent treatment of locally advanced disease.

Although of potential interest in the field, the manuscript suffers from important flaws. In particular, relevant aspects for immunotherapeutic treatment such as the intra-tumour heterogeneity, the tumour mutation burden, the generation of immunogenic neoantigens and the composition and immunosuppressive role of tumour microenvironment were not adequately addressed or totally ignored.

It is stated that: "...even standard cytotoxic treatment has an impact on immune TME composition, influencing long-term prognosis in some tumors, such as pancreatic, rectal, and even EC"  but no explanation/description of the underlying mechanisms was given. This is of relevance considering the frequent combination of radio/chemiotherapy with immunotherapeutic treatments.

The mechanism underlying the enhanced immunogenicity of microsatellite unstable tumours was not clearly explained. Immunotherapeutic approaches other than immune checkpoint inhibitors were limited to cancer vaccines and TIL-based immunotherapies. No description was provided of other novel strategies currently explored such as immunotherapies exploiting CAR T cells or bispecific T-cell engagers.

The results section is mainly a list of completed or ongoing trials with only little interpretation. The discussion is very short with no consideration of the challenges and perspectives of combining immunotherapy with standard treatments.

There are several conceptual errors along the text that should be corrected.

e.g.: line 127: Microsatellite instability (MSI), a condition of genetic hypermutability resulting from 127 DNA mismatch repair proteins (dMMR) 

Author Response

Reviewer 1

The review manuscript is aimed at describing the rationale for immunotherapy in different stages of esophageal cancer treatment, with a particular accent on curative intent treatment of locally advanced disease.

Although of potential interest in the field, the manuscript suffers from important flaws. In particular, relevant aspects for immunotherapeutic treatment such as the intra-tumor heterogeneity, the tumor mutation burden, the generation of immunogenic neoantigens and the composition and immunosuppressive role of tumor microenvironment were not adequately addressed or totally ignored.

We thank the reviewer for this comment. While keeping the paper oriented to clinicians and not to basic researchers, we thoroughly revised the discussion section to provide relevant information about the context tumor biology, and the mechanism of different immunotherapeutic treatments. It is indeed of great value to help the readership understand the underlying mechanisms and theoretical context of the subject treated in our manuscript.

It is stated that: "...even standard cytotoxic treatment has an impact on immune TME composition, influencing long-term prognosis in some tumors, such as pancreatic, rectal, and even EC"  but no explanation/description of the underlying mechanisms was given. This is of relevance considering the frequent combination of radio/chemiotherapy with immunotherapeutic treatments.

Indeed, as it is a very relevant clinical issue, the mechanisms of interaction between radio/chemotherapy and the tumor microenvironment and immune response have been now extensively developed in the discussion section.

The mechanism underlying the enhanced immunogenicity of microsatellite unstable tumors was not clearly explained. Immunotherapeutic approaches other than immune checkpoint inhibitors were limited to cancer vaccines and TIL-based immunotherapies. No description was provided of other novel strategies currently explored such as immunotherapies exploiting CAR T cells or bispecific T-cell engagers.

Several molecular biology aspects have been extensively developed, to address TIL-based immunotherapies, different types of cancer vaccines and CAR T cell treatment (discussion). The Microsatellite instability section is meant to explain why in this specific context (MSI-h lesions) immunotherapy is a particularly valuable treatment choice, without further details on the genetic disorder of micro-satellite instability which is not the subject of the study. We strongly feel that the revised version provides a robust and detailed overview of the theoretical background of immunotherapy, without going into excessive details on pathophysiology and experimental treatments, which might blur the manuscript’s message and confuse readers.

The results section is mainly a list of completed or ongoing trials with only little interpretation. The discussion is very short with no consideration of the challenges and perspectives of combining immunotherapy with standard treatments.

We thank the reviewer for this remark. As per the usual manuscript structure, the results section is dedicated to presenting relevant studies, and has intentionally been left without interpretation. The discussion has been extensively revised to provide interpretation of immunotherapy trials and their outcomes, while the very important subject of future challenges and perspectives of immunotherapy has also been added.

There are several conceptual errors along the text that should be corrected e.g.: line 127: Microsatellite instability (MSI), a condition of genetic hypermutability resulting from DNA mismatch repair proteins (dMMR) 

We thank the reviewer for the comment, the phrase has been corrected in the manuscript to give a clearer message.

Reviewer 2 Report

1. Page 4, lane 186, there is a period mark after the word "while". It should be a typo.
2. Are there any ongoing clinical trials in advanced/metastatic settings?
3. Are there any CAR-T immunotherapy studies in EC?
4. Overall, the clinical studies of EC immunotherapy were broadly reviewed, but the manuscript was kinda flat and dull. Further conclusion/discussion is necessary. What are the challenges of immunotherapy in EC? Questions readers may be interested in, such as, are there any biomarkers to predict treatment efficacy and patient response? What is the possible mechanism for the development of resistance to cancer immunotherapy? How does chemotherapy/radiotherapy affect immunotherapy?

Author Response

  1. Page 4, lane 186, there is a period mark after the word "while". It should be a typo.

Thank you for this remark. We corrected it.

  1. Are there any ongoing clinical trials in advanced/metastatic settings?

Several trials are ongoing in the metastatic setting; however, the manuscript is focused on providing current evidence in the curative context and earlier lines of treatment. As immunotherapy has long been used metastatic and heavily pre-treated patients, we strongly feel that adding this section in the current manuscript would represent an overwhelming volume of data with little or no new implication in current practice.

  1. Are there any CAR-T immunotherapy studies in EC?

The discussion section has been amended to discuss CAR-T immunotherapy; however, we were not able to retrieve major CAR-T trials specifically for EC.

  1. Overall, the clinical studies of EC immunotherapy were broadly reviewed, but the manuscript was kind of flat and dull. Further conclusion/discussion is necessary. What are the challenges of immunotherapy in EC? Questions readers may be interested in, such as, are there any biomarkers to predict treatment efficacy and patient response? What is the possible mechanism for the development of resistance to cancer immunotherapy? How does chemotherapy/radiotherapy affect immunotherapy?

We thank the reviewer for this comment and suggestion. We performed accordingly an extensive revision of the discussion section to treat in a more comprehensive manner some relevant issues on our study’s subject. Immunotherapy is a vast field of knowledge, and no single study can be expected to cover all relevant subjects. However, we strongly feel that with having addressed the reviewers’ remarks, we have been able to provide a detailed overview on the current status of immunotherapy, useful to all related clinicians.

Reviewer 3 Report

The review by Farinha et al. is well designed, organized and presented.

There are some minor spelling issues (typo 178, a promising 186, while. tumor load 186)

While the results are extensively and nicely presented, I would expect a more extensive and critical discussion for this important review article.

2 important issues are highlighted (many studies in asian populations only, high financial burden of immunotherapy for most health systems), but not at all brought into a larger context. Are the results transferable to a western population or how do we design future immunotherapy trials for the western world. I can only guess that ethical reasons play a role why most trials happen in asia-please discuss! How expensive is a typical immunotherapy that will most likely come into clinical practice soon (after the 577 study)? Is it realistic that thousands of dollars will be invested by health sytems into a drug that is introduced into clinical practice after a study that was sponsored by a pharmaceutical company? Please discuss extensively!

Also more in general: Most of the cited and listed ongoing studies are probably not investigator-initiated. How can we rule out a massive bias by pharmaceutical companies and how can we as physicians guarantee access to modern treatment strategies in the future?

Author Response

There are some minor spelling issues (typo 178, a promising 186, while. tumor load 186)

Thank you for these remarks. We rectified the errors.

While the results are extensively and nicely presented, I would expect a more extensive and critical discussion for this important review article.

2 important issues are highlighted (many studies in Asian populations only, high financial burden of immunotherapy for most health systems), but not at all brought into a larger context. Are the results transferable to a western population or how do we design future immunotherapy trials for the western world. I can only guess that ethical reasons play a role why most trials happen in asia-please discuss! How expensive is a typical immunotherapy that will most likely come into clinical practice soon (after the 577 study)? Is it realistic that thousands of dollars will be invested by health systems into a drug that is introduced into clinical practice after a study that was sponsored by a pharmaceutical company? Please discuss extensively!

Also, more in general: Most of the cited and listed ongoing studies are probably not investigator initiated. How can we rule out a massive bias by pharmaceutical companies and how can we as physicians guarantee access to modern treatment strategies in the future?

We thank the reviewer for the encouraging comments and excellent points raised. They have been extensively addressed in the discussion section.

Round 2

Reviewer 3 Report

The review has been nicely improved. Congratulations to this nice overview of the state of the art.

Author Response

Thank you very much!